# A Method to Explore the Best Mixed-Effects Model in a Data-Driven Manner with Multiprocessing: Applications in Public Health Research

Hyemin Han 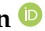

Educational Psychology Program, University of Alabama, Tuscaloosa, AL 35487, USA; hyemin.han@ua.edu;
Tel.: +1-205-348-0746

**Abstract:** In the present study, I developed and tested an R module to explore the best models within the context of multilevel modeling in research in public health. The module that I developed, *explore.models,* compares all possible candidate models generated from a set of candidate predictors with information criteria, Akaike information criterion (AIC), and Bayesian information criterion (BIC), with multiprocessing. For testing, I ran *explore.models* with datasets analyzed in three previous studies in public health, which assumed candidate models with different degrees of model complexity. These three studies examined the predictors of psychological well-being, compliance with preventive measures, and vaccine intent during the COVID-19 pandemic. After conducting model exploration with *explore.models*, I cross-validated the nomination results with calculated model Bayes Factors to examine whether the model exploration was performed accurately. The results suggest that *explore.models* using AIC and BIC can nominate best candidate models and such nomination outcomes are supported by the calculated model Bayes Factors. In particular, all the identified models are superior to the full models in terms of model Bayes Factors. Also, by employing AIC and BIC with multiprocessing, *explore.models* requires a shorter processing time than model Bayes Factor calculations. These results indicate that *explore.models* is a reliable, valid, and feasible tool to conduct data-driven model exploration with datasets collected from multiple groups in research on health psychology and education.

**Keywords:** data science; data-driven analysis; model exploration; mixed-effects model; public health

## 1. Introduction

Researchers in various fields involving data collection and analysis across multiple heterogeneous groups have employed mixed-effects models including fixed and random effects. The mixed-effects model method enables us to examine associations between predictors and the dependent variable of interest at the population level (fixed effects). It also allows us to assess how the intercepts (random intercepts) and the associations as mentioned earlier vary across different groups (random slopes), especially when observations are nested within groups [1]. Usually, the fixed effects can be understood as slopes in regression that are supposed to be common across different groups. The random intercepts are about whether each group has a different outcome variable mean. If there are significant random intercepts, then the intercept of each group should be adjusted in regression. Finally, random slopes indicate whether slopes significantly vary across different groups. If there are significant random slopes, a slope in a specific group is calculated by both fixed effects (global slopes) and random effects (group-specific slope adjustments). When data are collected from multiple groups, failing to consider group-level factors in the analysis can lead to misleading results [2]. For example, ignoring the potential random effects at the group level can inflate false positives, possibly resulting in overconfident estimates [3].

For instance, we may consider a global public health study, conducted across 43 countries by global public health researchers, which explored the relationship between people's

trust in government and science and COVID-19 vaccine intent across 43 countries [4]. We can straightforwardly predict that trust would positively predict intent to receive COVID-19 vaccines [5]. However, without considering random effects in such a case, the estimated regression coefficients of fixed effects are likely to be biased. First, the baseline level of vaccine intent may significantly vary across different countries, while the assuming predictors are the same [6]; this possibility warrants the addition of random intercepts to the analysis model. Second, due to different political situations and cultural backgrounds in various countries, the relationship between trust in government and science and vaccine intent may also significantly vary across countries [7]. This suggests that we should also consider random slopes. The analysis results in the study demonstrated that the regression models, including random intercepts and slopes, significantly predicted outcome variables better than the simpler models that only possessed fixed effects [4].

Furthermore, many health psychology and education researchers interested in investigating their research questions in multiple countries or contexts have widely employed mixed-effect analysis methods. For example, several studies that used this approach were published in the *European Journal of Investigation in Health, Psychology and Education*. Nasvytienė and Lazdauskas [8], Ta et al. [9], and Lochbaum and Sisneros [10] included both fixed effects and random effects in their analysis models to examine the relationship between candidate predictors and outcome variables across different conditions and contexts in the fields of health, psychology, and education. These papers demonstrated that mixed-effect analysis methods have been frequently used in health psychology and education beyond COVID-19-related research.

Let us assume that researchers aim to conduct a data-driven analysis to uncover optimal models predicting dependent variables of interest within such studies involving various heterogeneous groups. In such a case, they should carefully consider what action to take next. Although it is not deemed methodologically appropriate, in many studies, researchers tend to test a full model, which includes all predictors of interest, and then examine which predictors are significant based on the resulting *p*-values [11]. This is not ideal since the false positives can be inflated when the full model is tested. Moreover, *p*-values per se can only suggest whether null hypotheses (i.e., coefficients are zero) shall be rejected instead of whether alternative hypotheses of our interest shall be accepted [12]. Also, by testing only one full model, epistemologically, researchers are deemed to test one hypothetical model based on their conceptual assumptions instead of performing data-driven model exploration [13]. In terms of model fitting, although full models are likely to possess the best predictive accuracy (e.g., the highest $R^2$), they tend to be overfitted to the dataset used for regression [14]. As a result, the full models may be unable to accurately predict reality beyond the analyzed data [15]. Thus, despite its wide use and simplicity, testing one full model via *p*-values should not be considered an appropriate data-driven approach for best model exploration.

If researchers are genuinely interested in searching for the model that best explains their data, they should employ genuine data-driven methods instead of delving into one full regression model. Researchers interested in data-driven analysis have developed and utilized several data-driven model exploration methods. In the following subsection, I will overview several existing methods for simple model exploration in the field, such as step-wise regression, Bayesian model exploration, Bayesian Model Averaging (BMA), and regularization.

### 1.1. Methods for Model Exploration

First, we may consider step-wise regression methods based on the frequentist perspective [16]. These methods allow researchers to find the best candidate model by adding predictors to a null model (forward selection) or removing them from a full model (backward selection) step-wise. The forward or backward variable selection processes are performed until a statistical indicator used for testing (e.g., *p*-value, model information criterion) reaches a certain threshold [17]. Although these methods are epistemologically better at

data-driven model exploration than full model testing, they have several limitations. First, the variable selection process can be arbitrary; for example, different step-wise methods may suggest different outcome models [17]. Thus, model arbitrariness and uncertainty can be problematic [18]. Moreover, from a practical perspective, in the case of mixed-effects analysis, step-wise methods are difficult to implement because it is necessary to deal with candidate predictors at different levels [19].

Second, it is possible to compare model Bayes Factors (BFs) across different candidate models [20]. During model selection and comparison, BFs represent the extent to which a specific hypothesis or model is superior to others [21]. They are calculated based on Bayes' Theorem:

$$P(H|D) = \frac{P(H)P(D|H)}{P(D)} \tag{1}$$

According to this theorem, the posterior distribution $P(H|D)$, which indicates the probability that a hypothetical model of interest is true given data, is calculated by updating the prior distribution $P(H)$, which indicates the likelihood of the hypothetical model at the beginning, with data [12]. BFs are then quantified in terms of the degree of the evidence that updated the probability [21]. For instance, when Model A is compared with Model B, $BF_{AB}$ indicates how strongly observed evidence more favorably supports Model A versus Model B. In such a case, the model $BF_{AB}$ can be calculated as follows:

$$BF_{AB} = \frac{P(H_A|D)}{P(H_B|D)} \frac{P(H_B)}{P(H_A)} \tag{2}$$

where $P(H_A)$ and $P(H_B)$ represent the prior distribution of the model likelihood of Models A and B, respectively, and $P(H_A|D)$ and $P(H_B|D)$ are the posterior distribution of the model likelihood of Models A and B, respectively. $BF_{AB}$ greater than one suggests that Model A is more favored than Model B by the data.

Compared with *p*-values, BFs are deemed more appropriate for data-driven model exploration since they directly quantify which model is superior to others when data are observed [22]. Also, BFs enable researchers to compare all possible candidate models, so unlike conventional step-wise approaches, BF-applied model exploration is free from concern regarding arbitrariness. However, several practical limitations warrant further consideration. First, compared with frequentist approaches, Bayesian analysis, particularly Bayesian multilevel modeling (MLM), requires a significantly longer period to complete the Bayesian posterior probability calculation through iterative updating [23]. Second, in the case of MLM, which is the main interest of this paper, the existing R libraries (e.g., *BayesFactor* implementing diverse model exploration with feasibility) only allow one to explore models with random intercepts but without random slopes [24].

Third, researchers may consider employing information criteria for model selection, such as the Akaike information criterion (AIC) and the Bayesian information criterion (BIC). These criteria are calculated with the log-likelihood value, which is assumed to be improved by adding additional predictors to a model [25]. As mentioned above, the problem in model exploration is that the unnecessary addition of predictors not supported by a theory or hypothesis can result in overfitting. Thus, merely relying on log-likelihood values per se in model exploration may be misleading [15]. To address this issue, AIC and BIC consider the number of predictors included in models for their calculations, so they penalize models with unnecessary predictors [26]. Such a penalization mechanism is well represented in the formulae used to calculate the criteria. The criteria are calculated as follows:

$$AIC = -2LL \times n_p \tag{3}$$

$$BIC = -2LL \times log(N) \times n_p \tag{4}$$

where $LL$ is a log-likelihood value, $n_p$ is the number of predictors, and $N$ is the number of observations. Generally, when multiple models are compared, those with lower AIC and BIC values are deemed superior to their competitors. Due to the presence of the $n_p$ term

as a multiplier, when two models predict a dependent variable of interest with the same accuracy, the model with fewer predictors is more favored than the other from the AIC's and BIC's perspectives.

In most cases, AIC prefers a model with more predictors than BIC [27]. Although both indicators intend to prevent overfitting by penalizing unnecessarily complex models, AIC is more interested in pursuing predictive accuracy by being more liberal. On the other hand, BIC is more concerned about consistency in model selection to minimize overfitting by being more stringent. When a sample size is sufficiently large, BIC is better at nominating the correct model when compared with AIC [27]. In addition, generally, among these indicators, BIC has been deemed a proxy for BF for model evaluation and comparison [26]. That said, compared with AIC and BIC, BFs can provide more direct and accurate grounds for model comparison despite their heavier computational calculations. Another point to consider is that some statisticians, particularly Weakliem [28], argue that BIC assumes a unit prior distribution that might deviate from the actual prior distribution of model probabilities in a specific research project. He urged researchers to calculate BFs with prior distributions that they carefully determined based on theoretical and empirical foundations [28].

Despite the limitations, I intend to test these indicators within the context of model exploration for MLM, due to several practical merits. First, although they are considered proxies for BFs, the processing time to calculate the criteria is significantly shorter than that to calculate exact BFs [26]. Because BF calculation requires iterations for posterior updating, when complicated multilevel models are examined, it may take more than hours or even days to complete MLM [23]. Second, at the very least, criteria can suggest directions for further model exploration while saving time, despite the critique that the unit prior distribution is assumed [29]. For instance, if we can identify a couple of candidate models via criteria, then we will be able to conduct Bayesian MLM with prior distributions more suitable for our research projects (e.g., Rouder and Morey [21]) with the identified candidate models. From a practical perspective, doing so will significantly save time compared with exploring candidate models with BFs.

There are several alternative approaches to generating plausible prediction models that are not completely suitable for identifying the best prediction models. First, we may consider BMA. BMA averages the most probable prediction models based on their Bayesian posterior probability [30]. Previous studies reported that BMA shows improved prediction accuracy, particularly cross-validation accuracy, and addresses uncertainty existing in model selection processes [14]. Second, variable selection and regularization methods, such as LASSO and elastic-net regression, can also be employed [31]. These methods are suitable for selecting variables and regularizing coefficients to minimize cross-validation errors in prediction and prevent potential overfitting [13,32]. Although these methods perform effectively in generating prediction models, compared with the model exploration method I will propose, they have several limitations when used in health and psychological research. The result from BMA does not suggest one specific best model; instead, it demonstrates coefficients from averaging multiple candidate models [33]. Furthermore, gathering information for statistical inference, such as significance, by performing regularization is more difficult than conventional analysis methods [34]. From the practical side, I could not find any available R packages implementing these methods within the context of mixed-effect analysis.

### 1.2. Current Study

Given the practical benefits of using information criteria in model exploration, I apply them when comparing candidate models within the context of MLM with feasibility. I developed and tested an R module to implement the automated exploration of the best prediction model among all possible candidate models with random intercepts and slopes. The exploration was conducted by comparing model information criteria, i.e., AIC and BIC, across all possible candidate models. I generated possible candidate models by creating all possible combinations of user-designated fixed effects, random intercepts, and random slopes. Because such exploration may involve numerous computational iterations, I employed

multiprocessing to maximize the utilization of computational resources and save time. Additionally, I conducted a Bayesian mixed-effects analysis with the nominated best models to check whether the developed model exploration method can appropriately identify the best candidate models. To examine how the tool can be applied in reality, I tested it with large-scale international datasets used in previous studies in public health [4,5,35].

Blackburn et al. [4] examined which factors predict one's intent to receive the COVID-19 vaccine. Ntontis et al. [35] tested predictors significantly associated with psychological well-being during the COVID-19 pandemic. Han [5] explored the relationship between trust in society and compliance with preventive measures during the pandemic. Given that all three datasets were collected across different countries, they are suitable for MLM.

I expect that the exploration and analysis of these studies by Blackburn et al., Han, and Ntontis et al., with the novel data-driven method will contribute to the methodology in theoretical and application research on health behavior. They all address the potential predictors of health-related behaviors within the context of a public health crisis, e.g., the COVID-19 pandemic, such as vaccine intent and compliance with preventive measures [6,36,37]. The abovementioned studies employed theories closely related to health behavior in psychology and public health, including, but not limited to, the theoretical framework for conformity and compliance [38] and the theory of anti-intellectualism and human behavior [39]. Hence, if we can better explore data within this topical area with the data-driven analysis method, then such a method will help behavioral researchers in public health better understand the underlying mechanisms of human behavior and explore the most plausible theoretical model best supported by data.

## 2. Methods

All R code and data files for model exploration and tutorials are available via GitHub: https://github.com/hyemin-han/Explore_Mixed_Models (accessed on 1 February 2024). Tests with three previous study datasets are also provided as tutorials under https://github.com/hyemin-han/Explore_Mixed_Models/tree/main/Tests (accessed on 1 February 2024). Users may consider modifying the tutorial codes for their own use.

### 2.1. Software

I composed a customized R code to conduct model exploration within the context of MLM. This R code requires the following libraries as dependencies: *foreach, parallel, doParallel*, and *lmerTest* [40,41]. This software consists of three parts. First, there is the generator of all possible candidate models. Second, it has the ability to calculate information criteria for all possible candidate models via multiprocessing. Finally, it includes the ability to sort resultant candidate models based on a specific information criterion designated by the user. First, model exploration is conducted by calling a function, *explore.models*. *explore.models* requires a data frame containing data to be used; an R formula including a dependent variable, all fixed effects, and control variables; a string specifying a group variable; a list of strings specifying random slopes; a list of strings specifying variables that must be included in all candidate models; and several cores to be used for multiprocessing. For instance, hypothetically, once we call

$$\textit{explore.models} \ (\text{data}, Y \sim X1 + X2 + X3, 'G', c('X1','X2'), 'X3', 4)$$

then it explores all possible candidate models based on this full model:

$$Y \sim X1 + X2 + X3 + (1 + X1 + X2|G) \tag{5}$$

The formula specifies that the full model includes three fixed effects, X1, X2, and X3, and two random slopes, X1 and X2, at the group level specified by a group variable, G. Given that the X3 is designated as a variable that must be included, all candidate models to be explored will have X3. While exploring the models, to save time, four cores are utilized for multiprocessing as specified by the last parameter.

To generate the list of all possible candidate models, *explore.models* then internally calls another function, generate.RS.comb, which is a candidate model generator. Similarly, generate.RS.comb requires an R formula including a dependent variable, all fixed effects, and control variables; a string specifying a group variable; a list of strings specifying random slopes; and a list of strings specifying variables that must be included in all candidate models. In the case mentioned previously, *explore.models* internally calls the generate.RS.comb function to create a list of candidate models as follows:

$$\text{generate.RS.comb } (Y \sim X1 + X2 + X3, \text{ 'G', } c(\text{'X1','X2'}), \text{ 'X3'})$$

Then, the resultant list of all 13 possible candidate models is as follows:

$Y \sim X3$.
$Y \sim X1 + X3$.
$Y \sim X2 + X3$.
$Y \sim X1 + X2 + X3$ (so far, models with only fixed effects).
$Y \sim X3 + (1|G)$.
$Y \sim X1 + X3 + (1|G)$.
$Y \sim X2 + X3 + (1|G)$.
$Y \sim X1 + X2 + X3 + (1|G)$ (so far, models including a random intercept).
$Y \sim X1 + X3 + (1+X1|G)$.
$Y \sim X2 + X3 + (1+X2|G)$.
$Y \sim X1 + X2 + X3 + (1+X1|G)$.
$Y \sim X1 + X2 + X3 + (1+X2|G)$.
$Y \sim X1 + X2 + X3 + (1+X1+X2|G)$ (so far, models including random slopes).

When calling *explore.models*, three parameters (i.e., the group variable, the list of random slopes, and the list of variables) are required. The number of cores being employed is optional (default = 1). When the list of random slopes is not provided, the function only explores fixed effects and random intercept models. When the group variable is not specified, the function examines only fixed effect models. When the number of cores is not specified, the exploration processes are performed only with one core without multiprocessing.

Once all candidate models are generated by *generate.RS.comb*, *explore.models* conducts MLM with *lmerTest* for all generated candidate models. For each model, three indicators, i.e., LL, AIC, and BIC, are calculated and stored for further examination. When all model exploration processes are completed, a data frame containing the formula of the tested model, LL, AIC, and BIC for every tested model is returned.

Finally, for the best model selection, the *sort.result* function sorts the resultant models with a designated criterion. It requires two input variables: a list containing the model exploration results generated by *explore.models* and a criterion to be used. If a user does not specify the criterion, the default criterion, BIC, is employed. Alternatively, either LLs or AICs can be used. Once *sort.result* is performed, it returns a list of sorted model exploration outcomes. With the *head* function provided by R, users can examine which models are deemed the best candidates in terms of the lower BIC or AIC or the higher LL.

### 2.2. Tested Datasets

To test the developed functionality, I employed datasets from three studies in public health [4,5,35]. The three studies examined the factors predicting psychological well-being and compliance among participants recruited from multiple countries during the COVID-19 pandemic. Given that the studies utilized large-scale multi-national datasets, I assumed that the datasets were appropriate to test *explore.models* designed for MLM. The original three studies also used Bayesian MLM to examine the associations between hypothesized predictors and dependent variables of interest. However, because a tool to explore multilevel models, such as *explore.models*, which I invented, did not exist when the studies were conducted, they merely compared the null model, fixed effects model, random intercept model, and random slope model while including all hypothesized predictors.

Hence, I decided to explore all possible candidate models in terms of possible combinations of predictors with the datasets. These three studies tested full models with different degrees of complexity: Blackburn et al. examined two predictors [4], Ntontis et al. examined four [35], and Han examined seven [5]. I intentionally employed these three studies with full hypothesized models with three different levels of model complexity to test the processing time involving *explore.models* across tests with various levels of model complexity.

First, Blackburn et al. examined how participants' conspiratorial beliefs and trust in governments predicted compliance with COVID-19 preventive measures, particularly vaccine intent. Their dataset was collected from 15,740 participants across 43 countries [4]. Among the three tested studies, Blackburn et al. employed the simplest hypothetical full model [4]:

$$Vaccine\ intent \sim Conspiratorial\ beliefs + Trust\ in\ governments + \textbf{Demographics}+ \\ (1 + Conspiratorial\ beliefs + Trust\ in\ governments | Country) \tag{6}$$

where the highlighted part, "Demographics," includes demographic variables that must be included in all candidate models, i.e., gender, age, educational level, employment status, relationship status, and socioeconomic status. One minor point for consideration is that this study did not examine one's religious belief as a predictor [42]. Given that some people may not want to get vaccinated due to their religious beliefs, even if they trust the effectiveness of vaccines, the analyzed dataset might not include all potential predictors comprehensively. Thus, this might need to be noted as a possible limitation of the analyzed dataset. Based on the full model, my R code generated 13 candidate models. All data and source code files used in this study are available to the public via GitHub, https://github.com/hyemin-han/COVIDiSTRESS2_Vaccine (accessed on 1 February 2024).

Second, Ntontis et al. tested how primary and secondary stressors, group identity, and perceived social support predicted perceived stress [35]. They analyzed data collected from 14,600 participants across 43 countries. The following is the full model hypothesized in this study:

$$Perceived\ stress \sim Primary\ stressors + Secondary\ stressors + Group\ identity+ \\ Perceived\ social\ support + \textbf{Demographics}+ \\ (1 + Primary\ stressors + Secondary\ stressors + Group\ identity + Perceived\ social\ support | Country) \tag{7}$$

where the highlighted part, "Demographics", includes demographic variables that must be included in all candidate models, i.e., gender and socioeconomic status. Given that the model consisted of four hypothesized predictors, the number of generated candidate models was 97, indicating that this model had a higher complexity than Blackburn et al.'s [4]. The full dataset and source codes are available to the public via GitHub, https://github.com/hyemin-han/COVIDiSTRESS2_Stress.

Third, to test the most complicated model, I employed the dataset used in Han [5]. This study was conducted with a dataset of 20,601 participants from 62 countries. In this study, the author examined the relationship between trust in seven different domains, i.e., parliament or the government (Trust 1), the police (Trust 2), the civic service (Trust 3), the health system (Trust 4), the WHO (Trust 5), the government's effort to handle Coronavirus (Trust 6), the scientific research community (Trust 7), and compliance with preventive measures. For the current test, I focused on compliance with the vaccination recommendation as a dependent variable of interest. The full hypothesized model in this study was as follows:

$$Compliance \sim Trust1 + Trust2 + Trust3 + Trust4 + Trust5 + Trust6 + Trust7 + \textbf{Demographics}+ \\ (1 + Trust1 + Trust2 + Trust3 + Trust4 + Trust5 + Trust6 + Trust7 | Country) \tag{8}$$

where the highlighted part, "Demographics", includes demographic variables that must be included in all candidate models, i.e., gender, age, and educational level. Due to the larger number of candidate predictors and the complexity of the hypothesized model, a total of 2315 candidate models were generated. The full dataset and source codes are available to the public via the Open Science Framework, https://doi.org/10.17605/OSF.IO/Y4KGH.

*2.3. Test Procedures*

I conducted Bayesian MLM with *brms* to test the functionality [43,44]. Bayesian MLM was performed with the best models identified in terms of the lowest BIC and AIC, the null model only including designated demographic variables, and the full model including all candidate predictors and random effects. After conducting the Bayesian MLM mentioned above, I calculated model BFs to examine which model was more strongly supported by evidence than its counterparts. I used the null model as a baseline for the BF calculation and comparison. As a result, the following three BFs were calculated per test: $BF_{AIC,0}$, the BF comparing the model with the lowest AIC and the null model; $BF_{BIC,0}$, the BF comparing the model with the lowest BIC and the null model; and $BF_{Full,0}$, the BF comparing the full model and the null model. I assumed that the model demonstrating the highest BF was the best model.

In addition to the BF-based examination, I also investigated the processing time. The processing time was analyzed to evaluate the performance of my model exploration method. I was interested in whether the current tool can explore all possible candidate models more quickly than Bayesian MLM-based exploration. First, I measured how long it took to complete model exploration with *explore.models*. Second, I also examined the processing time required to complete the four Bayesian MLMs mentioned above. Because Bayesian MLM requires an extremely long time, only the processing time to complete the four Bayesian MLMs was used for comparison. The current model explorations and Bayesian MLMs were performed on Apple's MacBook Pro 16-inch with Apple M1 Pro (2021 edition), 16 GB memory, and macOS Monterey Version 12.6.6; in all cases, four cores were employed for multiprocessing.

## 3. Results

*3.1. Model Exploration Test Result*

Table 1 demonstrates the result of model exploration with three datasets. I conducted Bayesian MLM with models suggested by *explore.models*, which were based on AIC and BIC, the full and null models. Then, I calculated model BFs. First, $BF_{AIC,0}$ indicates the extent to which the model with the best AIC was more strongly supported by evidence than the null model. Second, $BF_{BIC,0}$ provides information about the extent to which the model with the best BIC was more favored by the data than the null model. Finally, $BF_{Full,0}$ was calculated to examine the extent to which the full model including all candidate predictors was more strongly supported by evidence than the null model. In the case of the simplest model exploration, Blackburn et al. [4], the model with the best AIC, and the model with the best BIC were identical to the full model, so only one model BF value was reported.

When Ntontis et al.'s study was examined [35], the model with the best AIC was identical to the full model. The model with the best BIC, which was uniquely recommended by *explore.models* and more stringent than the model with the best AIC and the full models, was best supported by evidence. The following is the nominated model with the best BIC:

$$Perceived\ stress \sim Primary\ stressors + Secondary\ stressors + Group\ identity+$$
$$Perceived\ social\ support + \textbf{Demographics} + (1 + Primary\ stressors|Country) \tag{9}$$

Unlike the model with the best AIC and the full models, the model with the best BIC only included one random slope, i.e., primary stressors.

In terms of BFs, the model with the best BIC nominated by *explore.models* was deemed superior to the model with the best AIC and the full models. According to the widely used guidelines to interpret BFs, $2logBF \geq 2$ indicates that the model of interest is significantly

and positively supported by evidence compared with its counterpart [45]. Hence, the difference in the 2logBF value, 2(10,123.59 − 10,121.52) = 4.14, suggests that the BIC model was better supported by evidence than the other models.

In the case of Han [5], *explore.models* nominated the models with the best BIC and AIC. These nominated models included fewer predictors than the full model, like in Ntontis et al. [35]. The model with the best BIC was

$$Compliance \sim Trust1 + Trust2 + Trust3 + Trust4 + Trust5 + Trust6 + Trust7 + \textbf{Demographics} + \\ (1 + Trust5 + Trust6 + Trust7 | Country) \tag{10}$$

The model with the best AIC was

$$Compliance \sim Trust1 + Trust2 + Trust3 + Trust4 + Trust5 + Trust6 + Trust7 + \textbf{Demographics} + \\ (1 + Trust1 + Trust4 + Trust5 + Trust6 + Trust7 | Country) \tag{11}$$

When model BFs were compared, the two nominated models reported BF values that were significantly higher than the full model's BF value. When the model with the best BIC was compared with the full model, the difference in the 2log(BF) was 2(4296.58 − 4248.95) = 95.26. When the model with the best AIC was compared, the difference in the 2log(BF) became 2(4297.25 − 4248.95) = 96.60. These 2logBF values even exceeded the threshold for the presence of very robust evidence, which was 10 [45]. Thus, it can be concluded that the models recommended by *explore.models* were significantly more strongly supported by evidence than the full model. Although the BF of the model with the best AIC was slightly higher than that of the model with the best BIC, the difference in 2logBF was 2(4297.25 − 4296.58) = 1.34 and non-significant. This was below the threshold for the presence of the positive evidence mentioned above. It may suggest that either the model with the best BIC or with the best AIC was not substantially superior to the other.

**Table 1.** Model exploration result.

|  | Blackburn et al. (2022) [4] | Ntontis et al. (2022) [35] | Han (2022) [5] |
|---|---|---|---|
| Model with best BIC |  | 10,123.59 | 4296.58 |
| Model with best AIC | 4250.08 | 10,121.52 | 4297.25 |
| Full model |  |  | 4248.95 |

Note. Numbers represent $log(BF_{M0})$.

### 3.2. Processing Time Analysis

Table 2 reports the outcome of the processing time analysis. As predicted, the processing time was positively associated with the complexity of the full hypothesized model (Blackburn et al. [4] < Ntontis et al. [35] < Han et al. [5]). Notably, when Blackburn et al.'s study was tested [4], the model nominated by the AIC and BIC was the full model, so only this model was compared with the null model with Bayesian MLM. In all cases, *explore.models* required a significantly shorter time to explore all possible candidate models. For instance, when Han was examined [5], Bayesian MLM with the simplest model, i.e., the null model, required 18.85 s to complete. When *explore.models* explored all 2315 candidate models, it took 3314.29 s. Even when one assumes that all 2315 candidate models are identical to the null model, it would take 18.85 s × 2315 models = 43,637.75 s to explore 2315 models with Bayesian MLM. All other candidate models were more complicated than the null model, so the actual processing time would be significantly longer than the estimate of 43,637.75 s.

**Table 2.** Processing time analysis.

| | | Blackburn et al. (2022) [4] | Ntontis et al. (2022) [35] | Han (2022) [5] |
|---|---|---|---|---|
| Number of candidate models | | 13 | 97 | 2315 |
| Elapsed time | *explore.models* | 3.06 s | 18.83 s | 3314.29 s |
| | BIC-best model | | 65.00 s | 438.67 s |
| | AIC-best model | 171.86 s | 153.74 s | 609.69 s |
| | Full model | | | 841.82 s |
| | Null model | 20.20 s | 15.59 s | 18.85 s |

## 4. Discussion

In this study, I developed and tested an R module to search for the best model within the context of MLM. The model exploration tool, i.e., *explore.models*, which I invented, allows users to explore the best prediction model among candidate models. The candidate models are generated by combining candidate predictors at the population and group levels, following the users' directions. Instead of step-wise regression, which is likely to produce arbitrary outcomes and inflate false positives [17,19], or BF-based model exploration, which is time-consuming [26], I utilized information criteria, AIC and BIC, as proxies for BFs in model exploration. Furthermore, multiprocessing was applied to reduce the processing time. The module was tested with large-scale international datasets analyzed in three previous studies in public health, namely by Blackburn et al., Han, and Ntontis et al. [4,5,35]. These three studies were employed to test the module with three multilevel models with different degrees of model complexity.

*explore.models* suggests that candidate models based on the best AIC and BIC are also supported by BFs calculated by Bayesian MLM with *brms*. In addition, when compared with Bayesian MLM, *explore.models* could explore and compare all possible candidate models, which ranged from 13 [4] to 2315 [5] models, within a significantly shorter processing time. Similar to the prediction of this study, *explore.models* was found to require a shorter processing time than Bayesian MLM to test all possible candidate models. The same trends were also found in the analyses of Blackburn et al. and Ntontis et al. [4,35]. The nominated models reported better BFs compared with the full models. Hence, the nominated stringent models are less susceptible to overfitting, as reported by better cross-validation results from previous studies (e.g., Han et al. [13,14]). Researchers who intend to conduct data-driven model exploration with multilevel models can use *explore.models* to save time while maintaining the credibility of the model selection process. The R codes include customized functions and tutorials available to the public via GitHub, meaning researchers can feasibly employ *explore.models* in their research projects.

Despite the practical benefits mentioned above, there are several limitations to the proposed method. First, as mentioned in the Introduction, information criteria are merely approximations of BFs [26], so they might not be able to provide full information about which model is superior to others [27]. For instance, BIC is deemed a proxy for BF with an assumption of the unit priors [28,29]. Although the unit priors assumption has been regarded as acceptable for model selection, as argued by Raftery [29], actual BFs may need to be calculated with proper prior distributions that are consistent with the data and research questions for ideal decision-making. This suggests researchers may start by quickly searching candidate models with *explore.models*, then conducting Bayesian MLM with the identified best candidate models (e.g., 1–5 best models) to calculate BFs and compare them. It could allow researchers to reasonably compromise between the computational complexity and credibility of model recommendations.

Second, although the developed tool could boost processing time by using information criteria and multiprocessing, it may still require a lengthy amount of time to complete

model exploration with a complex model. As tested with the three study datasets, the number of candidate models and processing time increased exponentially as the number of candidate predictors increased. When 13 models with two predictors were tested, 3.06 s was required. When the more complicated model, which included 97 candidate models with four predictors, was tested, it took 18.83 s to complete the exploration. Finally, when 2315 candidate models with seven predictors were tested, 3314.29 s was required to complete the task. Hence, if researchers intend to explore complicated candidate models, they should consider setting more restrictions, such as specifying more "should be included" variables or fewer candidate random slopes. Alternatively, researchers may employ large-scale cluster computing to increase the number of cores for calculation [23]. Han [23] demonstrated that utilizing 16 cores, which is feasibly performed via available computer clusters, such as Amazon AWS, decreased processing time by more than 90%. Given that I employed only four cores in the present study, cluster computing will effectively reduce processing time even when examining complex models and make the current method more feasible for researchers with limited time and resources.

Third, even though *explore.models* is supposed to nominate candidate models best supported by the provided data, the models do not necessarily produce theoretically and conceptually relevant and meaningful results [46]. That said, merely relying on this data-driven method may result in the nomination and employment of models without any theoretical or conceptual support, simply due to the apparent verisimilitude of the models. Instead, researchers may use the nominated candidate models as starting points for further investigations [26]. Researchers will also need to consider the theoretical implications of nominated models carefully before their use [46].

Given its nature and limitations, researchers who intend to employ *explore.models* should consider several points to prevent encountering spurious outcomes or spending an unnecessarily long time on this phase of research. First, before getting started, they must consider determining candidate predictors to be investigated and analyzed based on the relevant theories and previous studies (e.g., variables selected by theories in public health in previous international surveys on COVID-19 [47,48]). By doing so, the potential *explore.models* results can be supported at the theoretical and conceptual levels. Also, by limiting the number of candidate predictors based on theories and prior studies, researchers can limit processing time. In fact, according to the philosophy of science, even data-driven observations are likely to be guided by prior theoretical frameworks, at least implicitly [49]. Second, instead of interpreting the nominated model as a final model for theory building, researchers may utilize the nominated model as a starting point for further investigations. The further investigation guided by the nominated model will support the model's validity with potential feedback from additional theoretical and conceptual considerations [50]. Finally, depending on the sample size and researchers' intent, they should decide whether they focus on the model with the best BIC or the best AIC [27]. For instance, they may examine whether or not predictive accuracy is more important than preventing overfitting. Ideally, they may consider comparing and interpreting several top-nominated models instead of choosing one single model [26]. Perhaps additional studies should be conducted to optimize the computational process and to develop guidelines to determine prior distributions and which criteria should be employed for model recommendation.

## 5. Conclusions

In the present study, I developed and tested an R tool for model exploration for mixed-effect analysis. The developed tool, i.e., *explore.models*, can generate a credible list of candidate models for further exploration while minimizing processing time. I shared the source codes and tutorial files via GitHub, so researchers in health psychology and education, particularly those who intend to conduct cross-cultural and cross-national research projects, can feasibly employ this tool for data-driven analysis in their research projects.

**Funding:** This research received no external funding.

**Institutional Review Board Statement:** Not applicable.

**Informed Consent Statement:** Not applicable.

**Data Availability Statement:** All R code and data files for model exploration and tutorials are available via GitHub: https://github.com/hyemin-han/Explore_Mixed_Models (accessed on 1 February 2024). Tests with three previous study datasets are also provided as tutorials under https://github.com/hyemin-han/Explore_Mixed_Models/tree/main/Tests (accessed on 1 February 2024). In the case of Han [5], the full dataset as well as source codes are available to the public via the Open Science Framework, https://doi.org/10.17605/OSF.IO/Y4KGH (accessed on 1 February 2024).

**Conflicts of Interest:** The author declares no conflicts of interest.

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
