# Peer review of "A Method to Explore the Best Mixed-Effects Model in a Data-Driven Manner with Multiprocessing: Applications in Public Health Research"

_ejihpe, doi:10.3390/ejihpe14050088_

Round 1

Reviewer 1 Report

Comments and Suggestions for Authors

This is a well-written manuscript. The only thing I would recommend is expanding the introduction to include other methods commonly used in model selection, such as model averaging. A more thorough overview of alternative methods and their limitations would strengthen the proposal for this method (as is done for stepwise selection algorithms, already)

Author Response

Dear Reviewer 1,

Thank you very much for your invaluable comments to improve my paper. You can find how to revised the manuscript as per your comments below:

Comment

This is a well-written manuscript. The only thing I would recommend is expanding the introduction to include other methods commonly used in model selection, such as model averaging. A more thorough overview of alternative methods and their limitations would strengthen the proposal for this method (as is done for stepwise selection algorithms, already)

Response

Thanks a lot for your suggestion. In the revised manuscript, I added additional alternative model search methods following your suggestion:

There are several alternative approaches to generating plausible prediction models that are not completely suitable for identifying the best prediction models. First, we may consider BMA. BMA averages the most probable prediction models based on their Bayesian posterior probability [30]. Previous studies reported that BMA shows improved prediction accuracy, particularly cross-validation accuracy, and addresses uncertainty existing in model selection processes [14]. Second, variable selection and regularization methods, such as LASSO and elastic-net regression, can also be employed [31]. These methods are suitable for selecting variables and regularizing coefficients to minimize cross-validation errors in prediction to prevent potential overfitting [13,32]. Although these methods perform effectively in generating prediction models, compared with the model exploration method I will propose, they have several limitations in being used in health and psychological research. The result from BMA does not suggest one specific best model; instead, it demonstrates coefficients from averaging multiple candidate models [33]. Furthermore, gathering information for statistical inference, such as significance, by performing regularization is more difficult than the conventional analysis methods [34]. From the practical side, I could not find any available R packages implementing these methods within the context of mixed-effect analysis. (pp. 4-5)

Reviewer 2 Report

Comments and Suggestions for Authors

This article is a revolutionary approach in improving the quantitative research method which has some important weaknesses already. The best credit is the showing of program codes and other useful materials for readers to trace and drill down the details. There are some minor issues which need to be solved:

1) It is not so common to use first-person writing style to describe the work done in academic world.

2) In line #191, why are the variable names the same for fixed effects and random slopes? It is quite confusion.

3) From lines #208-210,  why is there no case for Y~X1 + X2? Same questions to line #213-214.

4) in line #223, why is no need to include 'number of cores'?

5) There are so many results mentioned. Should the author show them using tables?

6) In line #412, why not consider quantum computing? Large-scale cluster computing is not cheap.

The writing style and the presentation method need to be revamped.

Author Response

Dear Reviewer 2,

Thank you very much for your invaluable comments to improve my paper. You can find how to revised the manuscript as per your comments below:

Comment 1.

1) It is not so common to use first-person writing style to describe the work done in academic world.

Response 1.

Thanks a lot for your comment about using the first-person writing style. I understand that such is not recommended in some field, in the field of psychology, the writing convention, the APA style, encourages using the first-person expression. The style guidelines suggest authors avoid to use the third-person expression or passive voice unnecessarily. For additional information, please refer to these documents:

https://owl.purdue.edu/owl/research_and_citation/apa6_style/apa_formatting_and_style_guide/apa_stylistics_basics.html

https://apastyle.apa.org/style-grammar-guidelines/grammar/first-person-pronouns

https://apastyle.apa.org/blog/first-person-myth

Comment 2.

2) In line #191, why are the variable names the same for fixed effects and random slopes? It is quite confusion.

Response 2.

I appreciate your comment about the further clarification. Because ordinary multilevel modeling requires including random slopes that were already employed as fixed effects, you saw some overlapping variable names between them. Users can designate which fixed effect variables are supposed to be included as random slopes.

Comment 3.

3) From lines #208-210,  why is there no case for Y~X1 + X2? Same questions to line #213-214.

Response 3.

Thanks for your question. Because 

explore.models (data, Y X1 + X2 + X3, ’G’, c(’X1’,’X2’), ’X3’, 4)

Requires X3 should be included all possible models, X3 appeared all possible combinations. So, "X1 + X2" did not appear in the results.

Comment 4.

4) in line #223, why is no need to include 'number of cores'?

Response 4.

Thanks for your comment. I revised the sentence accordingly:

When calling explore.models, three parameters, the group variable, the list of random slopes, the list of variables, are required. The number of cores to be employed is optional (default = 1). (p. 6)

Comment 5.

5) There are so many results mentioned. Should the author show them using tables?

Response 5.

I appreciate your question. Because many numbers, including the model complexity and processing time, are required to be reported in the results section, tables were added despite the presence of many representative numbers appearing in the main text.

Comment 6.

6) In line #412, why not consider quantum computing? Large-scale cluster computing is not cheap.

The writing style and the presentation method need to be revamped.

Response 6.

Thanks for your suggestion. However, I decided not to add information about quantum computing, because quantum computing is deemed to be not ideal for exploring candidate models with big data while addressing classical problems. Please refer to this document for further details about the discussion:

https://spectrum.ieee.org/quantum-computing-skeptics

Reviewer 3 Report

Comments and Suggestions for Authors

Abstract

The abstract introduces a creative new model for multiprocessing public health research in a timely manner. Additionally, the abstract provides a detailed outline of how this model was created and acknowledges that it can be used for future research. However, this abstract is not appropriate for publication in its current state and requires major revisions throughout the paragraph. The abstract has major grammatical errors that must be addressed. There is an evident language barrier, which is noticeable throughout the abstract. The addition of another author with English as a first language could improve the flow of the article. Additionally, the abstract provides a detailed explanation of the methods, and could benefit from more background information, or conclusions that the author developed from their module’s data. It is difficult to understand what the author developed, what makes the author’s model different from existing models, and what the implications of this new model are from this abstract.

Line 1: explore the best models

Line 3: compared; the sentence is a run-on sentence, does not flow grammatically and hard to understand based on how it is currently written

Line 4: Provide abbreviations for Akaike Information Criterion (AIC) and Bayesian Information Criterion (BIC), they are used later in the abstract but never defined.

Line 5: Can we elaborate more on the three previous studies in public health? More information on these is needed. From its current state, the abstract does not have any novelty because what this R module is being compared to is unknown or not understood by the reader

Line 7-8: “After conducting model exploration with explore.models, I calculated the model Bayes Factors of the nominated best models for validation.” This whole sentence needs to be rewritten, unclear to readers, not easy to comprehend what the author is trying to say here.

Line 8-9: “The results suggested that explore.models using AIC and BIC was able to nominate best candidate models that also demonstrated superior model Bayes Factors compared with competitors, the full models in particular.” This whole sentence is grammatically incorrect, unreadable, and appears to be written by someone who does not use English as first language. I am unable to understand what the result is when given this sentence. Consider a complete revision, “The results suggested that explore.models using AIC and BIC was able to nominate best successfully identified full candidate models that also demonstrated superior model Bayes Factors when compared with competitors. the full models in particular

Line 11-12: explore.models required the shorter processing time compared with in comparison to complete model Bayes Factor calculations.

Line 12: “I discussed the implications of this R module for future research in the field.” This needs additional information. What are the implications? This is not an appropriate summary of the results or future directions.

Introduction

The introduction includes interesting data from a study regarding COVID-19 vaccination intent across many countries and suggests ways to improve the analysis of this data through more sophisticated models. The author explains that there is a gap for their style of models, identifying other literature that supports the development of their model. Also, the author gives a thorough explanation of the reasoning behind why they designed their study the way they did. But this may be more of a beneficial addition to the Methods section. Despite defining the gap and having some interesting data, major edits must be made to this introduction. One important detail that is missing from this introduction is the objectives of this article and what the researchers’ goals are. A lack of objectives gives the article a lack of purpose. Another major flaw of the introduction is that it requires major grammatical revisions. The introduction is hard to follow due to the abundance of run-on sentences and wordy explanations. Additionally, many different concepts are introduced but not thoroughly explained, as if there is an assumption that the reader is already an expert in this subject. Many of the studies and ideas mentioned would benefit from further explanation from the author, so the reader can comprehend why these are relevant to the introduction and the article.

Line 14: Why is there a 1. in front of introduction? There are not any other sections that are numbered, suggest removing this.

Line 17-20: This is a very long, run-on sentence. Consider breaking up into two sentences, for example. “The mixed-effects model method enables us to examine associations between predictors and the dependent variable of interest at the population level (fixed effects). It also allows us to assess how the intercepts (random intercepts) and the aforementioned associations vary across different groups (random slopes), especially when observations are nested within groups [1]."

Line 18: “(fixed effects)” what does it mean?

Line 19: “(random intercepts)” what does it mean?

Line 20: “(random slopes)” what does it mean?

Line 21-23: “in analysis, analysis” Consider revision of this entire sentence, putting the same word twice is hard for readers to comprehend. Consider the following revision: When data exists in multiple groups, failing to consider group-level factors in analysis can lead to misleading results.

Line 24: so likely resulting in to end up overconfident estimates

Line 25: For instance, we may consider A global public health study across 43 countries conducted in the field of global public health,

Line 26: which explored the relationship between people’s trust in government and science, and

Line 27: COVID-19 vaccine intent across 43 countries [ 4].

Line 28: Replace “get” with a more professional word, such as receive.

Line 30: to be biased.

Line 31: countries, while assuming predictors are the same [6], this possibility warrants warranting the necessity

Line 33: in different across various countries. Consider using another word since “different” was utilized in the Line above.

Line 35: Adding a semi colon makes this sentence too long. Consider ending the sentence in Line 34 and starting the next sentence: “This suggests that…” In fact, the analysis results in

Line 36: regression models, including random intercepts and slopes,

Line 37: significantly better predicted outcome variables significantly better in comparison compared with the simpler models that only

Line 38: possessed with fixed effects [4].

Line 41: “what shall we be supposed to do?” does not make grammatical sense, must be rewritten. It is an interesting way to propose a question in the article to make readers guess what comes next. Consider “we should consider what actions to take next

Line 43: examined

Line 44-47: This is a very long sentence that is hard to follow. Consider breaking up into two sentences for better flow for readers.

Line 52: might not be able may be unable to accurately predict

Line 53: data accurately [12].

Line 56: Then, If researchers are genuinely interested in searching for the model that best explaining explains

Line 58-59: Is the intent of this sentence to lead into the following paragraph, Methods for Model Exploration? If not, consider elaborating on these methods and employed, because this sentence seems like an unrelated ending to the paragraph.

Line 62: researchers or the researcher

Line 68-70: Consider combining these two sentences and removing such as forward and backward selection. “First, the variable selection process can be arbitrary, for example, different stepwise methods may suggest different outcome models”

Line 90: to others when given data

Line 93-94: This is not a complete sentence “However, there are several practical limitations warrant further considerations.” Consider adding limitations that warrant                                                    

Lines 96-99: Second, in the case of MLM, which is the main interest of this paper, the existing R libraries, such as BayesFactor, implementing diverse model exploration with feasibility only allowing exploring models with random intercepts, but without random slopes [21].

Line 101: Abbreviations for AIC and BIC should be introduced earlier as they are utilized prior to being defined here.

Line 103: above-mentioned should be edited, for example to mentioned above

Line 117-121: This is a very long sentence, consider breaking into multiple sentences for better flow for the reader

Line 122: In addition, in general,

Line 124: Compared with what criteria? What is the criteria?

Line 125: “despite their calculation is computationally heavier.” is grammatically incorrect. Consider revision, for example despite their heavier computational calculation

Line 126: Some argued for BIC must be clarified. Who argued for BIC? What was the basis for this argument? More detail is necessary.

Line 135: time,

Line 150: particularly when a complex model is examined Consider removing to improve flow of sentence

Line 157-158: Author refers to Blackburn et al., Han, and Ntontis et al. as if they have been introduced to the reader, but they have not

Line 162: Overuse of the word abovementioned, consider utilizing another word

Methods

The author provides tutorials in a working link at the beginning of the section so readers can utilize the codes for themselves. Also, the author provides very detailed lists and explanations of possible candidate models, while walking the reader step-by-step through their methods. The author addresses a gap in literature with an explanation of how they created this novel model with a detailed tutorial. However, there are major grammatical errors that persist into the methods section and take away from the author’s hard work. These run-on sentences and grammatical errors must be revised in order to allow readers to fully understand the methods section. In conclusion, the reader is unable to follow the methods to replicate the author’s model because the methods section is grammatically flawed.

Line 182-185: Consider revising the structure of this sentence, utilization of semi-colons is excessive

Line 222-224: “When calling explore.models, four parameters, the group variable, the list of random slopes, the list of variables that must be included, and the number of cores to be are not required.” Are all of these elements not required? Or just the number of cores? Consider clarifying and revising this sentence

Line 275: models was 97, indicating that

Line 288: Consider removing (7)

Line 304-305: Consider removing “as well.” It is repetitive after starting the sentence with “In addition to”

Line 305-306: Must consider rewriting the sentence “The processing time was analyzed to examine whether my model exploration method can complete comparing all possible candidate models more quickly than Bayesian MLMs”

Line 308: above-mentioned… We must be consistent with how we are spelling this word, there are times were it is spelled abovementioned and here it is hyphenated. Please edit for consistency

Line 309: Because Bayesian MLMs requires an extremely long time

Results

The results section contains major flaws. It is recommended to refrain from using I (first person) in scholarly writing. It was noted several times throughout this section. Additionally, the results section should solely focus on the results. The interpretation of the results and their significance should go in the discussion section. Please find the edits for this section below.

Line 316: Instead of I conducted, it is recommended to say  “this research was conducted with”

Line 317: “with explore.models with models suggested by explore.models”  Using with twice in a row sounds redundant,  also said models frequently in this sentence

Line 326-327: “Only included one random slope, primary stressors” It is suggested to say “only included one random slope, which was primary stressors”

Line 332: Instead of suggesting , use “suggests”

Line 335: It is recommend to say which was similar to after the word predictors

Line 337: There should not be a comma after the and, also suggest deleting the word and

Line 339: “When the BIC-best model as” Use was instead of as

Line 343: “,10” It is recommended to say which was 10

Line 343: “Thus, I shall conclude that the models” àIt can be concluded that the models”

Line 346: “was not significant, 2(4297.25 − 4296.58) = 1.34” It is suggested to say was 2(4297.25 − 4296.58) = 1.34 and not significant

Line 347: Instead of which, start new sentence with this was below

Line 351: Instead of saying as expected, say “as predicted”

Line 353: “One note is that when” it is recommended to say of note, when Blackburn

Line 359: “Even if we assume” it is recommended to say even if one assumes

Line 362: “,the actual processing time would be” It is suggested to say this means the actual processing time

Line 363: “estimate, 43637.75 seconds” instead of the comma, say estimate of 43637.75 seconds

Line 363: Do not say as expected, it is suggested to say “similar to the predictions of this study”

Line 364: Suggest taking out “the” before the word shorter

Line 365: “The same trends was also found from my analysis of” Instead of was, use were. Also instead of saying my, say “found from the analysis of”

Line 363-366: This part of the results likely belongs in the discussion section.

Discussion:

The discussion succeeded in comparing the model to other models and explaining the use for the model. The limitations were noted well with specific examples. Since the paper did not include a conclusion, some of the text from the discussion section should be added to the conclusion. The conclusion should restate the main points and discuss next steps. A weakness of this paper was the lack of further direction for future studies. Please find the edits for this section below.

Line 369-371: “explore.models, which I invented, allows its users to explore the best prediction model among candidate models; the candidate models are generated by the combination of candidate predictors at the population and group levels following the users’ directions.”

·       The sentence should not start with Explore.models

·       Which I inventedà It is recommended to say “invented by the author”, want to refrain from using first person

·       Instead of a semicolon highlighted in red, it is suggested to put a period and make the second part a different sentence.

Line 373: outcomes and to inflate false positivesà It is suggested to delete the word “to”

Line 380: Should the E be capital in explore.Models because it is the start of a sentence?

Line 381: What is brms? It is not defined anywhere else in the paper.  

Line 385: “Stringent models are less likely susceptible” Recommend saying either “less susceptible” or “less likely to be susceptible”

Line 387-389: “In general, I found that researchers who intend to conduct data-driven model exploration with multilevel models will be able to employ explore.models to save their time while maintaining the credibility of the model selection process.”

·       Rewording suggestion: Researchers who intend to conduct data-driven model exploration with multilevel models can use explore.models to save time, while also maintaining credibility of the model selection process.

Lines 389-390: “Also, since I composed the R codes, which include the customized functions, and tutorials available to the public via GitHub, researchers will be able to employ explore.models in their research projects feasibly.”

·       Rewording suggestion: The R codes include customized functions and tutorials available to the public via GitHub, meaning researchers will be able to feasibly employ explore.models in their research projects.

Line 393: “information criteria are” Say “is” in the place of “are”

Line 397: “been regarded practically acceptable” Rewording suggestion: Has been regarded as acceptable

Line 398: “with proper prior distributions coherent with data” Consider rewording

Line 399: Recommend changing “that suggests” to “this suggests”

Line 402-403: “It might allow researchers to compromise between computational complexity and credibility of model recommendation reasonably.” Suggest changing to: It could allow researchers to reasonably compromise between computational complexity and credibility of model recommendation.

Line 404-406: “Second, although I was able to boost the processing time via use of information criteria and multiprocessing, it still requires a long time to complete model exploration with a complex model.” Rewording suggestion: Although this study was able to boost the processing time via use of information criteria and multiprocessing, it still requires a lengthy amount of time to complete model exploration with a complex model.

Line 407-408: “13 models with 2  predictors → 97 models with 4 predictors → 2315 models with 7 predictors” Recommend writing this out in a sentence instead of using arrows.          

Line 409: “3.06 seconds → 18.83 seconds 409 → 3314.29 seconds” Recommend writing this out in a sentence instead of using arrows.          

Line 411-412: “setting more restrictions (e.g., specifying more "should be 411 included" variables or less candidate random slopes” It is suggested to say “setting more restrictions, such as”

Line 414: “ Third, although” It is recommended to say Even though instead

Line 415-416:the models do not necessarily provide theoretically and conceptually relevant and meaningful results” Rewording suggestion: Theoretically and conceptually relevant results that are meaningful.

Line 418: Possibly à This word can be deleted

Line 420: “Also, researchers will need to” à Rewording suggestion: Researchers will also need to consider

Line 423-424: “spending an unnecessarily long time for computation.” Rewording suggestion: spending an excessive amount of time computing the data

Line 424: “they may consider” It is suggested to say “they must consider”

Line 426: Instead of (e.g.,) Recommend making this a separate sentence and start it with for example

Line 430: Used also twice in this sentence, suggest deleting the also at the end of the sentence

Line 430: Apparently dataà Is this a term used in the field? The sentence read weird due to the word apparently.

Line 439: Do not start a sentence of with of course, you could delete this and start with the word ideally.

Conclusion: This article did not have a conclusion section; author should add one at the end of the discussion section.

Comments on the Quality of English Language

Line 316: Instead of I conducted, it is recommended to say  “this research was conducted with”

Line 317: “with explore.models with models suggested by explore.models”  Using with twice in a row sounds redundant,  also said models frequently in this sentence

Line 326-327: “Only included one random slope, primary stressors” It is suggested to say “only included one random slope, which was primary stressors”

Line 332: Instead of suggesting , use “suggests”

Line 335: It is recommend to say which was similar to after the word predictors

Line 337: There should not be a comma after the and, also suggest deleting the word and

Line 339: “When the BIC-best model as” Use was instead of as

Line 343: “,10” It is recommended to say which was 10

Line 343: “Thus, I shall conclude that the models” àIt can be concluded that the models”

Line 346: “was not significant, 2(4297.25 − 4296.58) = 1.34” It is suggested to say was 2(4297.25 − 4296.58) = 1.34 and not significant

Line 347: Instead of which, start new sentence with this was below

Line 351: Instead of saying as expected, say “as predicted”

Line 353: “One note is that when” it is recommended to say of note, when Blackburn

Line 359: “Even if we assume” it is recommended to say even if one assumes

Line 362: “,the actual processing time would be” It is suggested to say this means the actual processing time

Line 363: “estimate, 43637.75 seconds” instead of the comma, say estimate of 43637.75 seconds

Line 363: Do not say as expected, it is suggested to say “similar to the predictions of this study”

Line 364: Suggest taking out “the” before the word shorter

Line 365: “The same trends was also found from my analysis of” Instead of was, use were. Also instead of saying my, say “found from the analysis of”

Author Response

Dear Reviewer 3,

Thank you very much for your invaluable comments to improve my paper. You can find how to revised the manuscript as per your comments below:

Comment 1.

The abstract introduces a creative new model for multiprocessing public health research in a timely manner. Additionally, the abstract provides a detailed outline of how this model was created and acknowledges that it can be used for future research. However, this abstract is not appropriate for publication in its current state and requires major revisions throughout the paragraph. The abstract has major grammatical errors that must be addressed. There is an evident language barrier, which is noticeable throughout the abstract. The addition of another author with English as a first language could improve the flow of the article. Additionally, the abstract provides a detailed explanation of the methods, and could benefit from more background information, or conclusions that the author developed from their module’s data. It is difficult to understand what the author developed, what makes the author’s model different from existing models, and what the implications of this new model are from this abstract.

Line 1: explore the best models

Line 3: compared; the sentence is a run-on sentence, does not flow grammatically and hard to understand based on how it is currently written

Response 1.

First of all, I sincerely appreciate your numerous comments and suggestions to improve my manuscript. I found that they are very extensive and informative. It is my first time to receive this number of constructive comments and suggestions during the peer-review process, I do not know how to express my deepest gratitude.

In general, I addressed the language issues that you raised throughout the manuscript. During the process, I employed a professional language editing and improvement tool for further language improvement. In the most cases, I accepted your suggestions.

Below, you would find my responses to your comments if I decided to revise the manuscript somehow different from how you suggested. Detailed comments are also added when additional information is required to be provided. Otherwise, I modified the manuscript according to your feedback, so you can find the modified parts from the revised manuscript.

Comment 2.

Line 4: Provide abbreviations for Akaike Information Criterion (AIC) and Bayesian Information Criterion (BIC), they are used later in the abstract but never defined.

Response 2.

Thanks for your comment for explaining the abbreviations. I added the definitions earlier in the abstract.

Comment 3.

Line 5: Can we elaborate more on the three previous studies in public health? More information on these is needed. From its current state, the abstract does not have any novelty because what this R module is being compared to is unknown or not understood by the reader

Response 3.

Thank you for your comment for the additional information for the studies. The information is now added to the revised manuscript:

These three studies examined the predictors of psychological well-being, compliance with preventive measures, and vaccine intent during the COVID-19 pandemic. (p. 1)

Comment 4.

Line 7-8: “After conducting model exploration with explore.models, I calculated the model Bayes Factors of the nominated best models for validation.” This whole sentence needs to be rewritten, unclear to readers, not easy to comprehend what the author is trying to say here.

Line 8-9: “The results suggested that explore.models using AIC and BIC was able to nominate best candidate models that also demonstrated superior model Bayes Factors compared with competitors, the full models in particular.” This whole sentence is grammatically incorrect, unreadable, and appears to be written by someone who does not use English as first language. I am unable to understand what the result is when given this sentence. Consider a complete revision, “The results suggested that explore.models using AIC and BIC was able to nominate best successfully identified full candidate models that also demonstrated superior model Bayes Factors when compared with competitors. the full models in particular”

Line 11-12: explore.models required the shorter processing time compared with in comparison to complete model Bayes Factor calculations.

Response 4.

I appreciate the comments for language improvements. I revised and rewrote the parts that you mentioned accordingly.

Comment 5.

Line 12: “I discussed the implications of this R module for future research in the field.” This needs additional information. What are the implications? This is not an appropriate summary of the results or future directions.

Response 5.

Thanks for your suggestion for additional information at the end of the abstract. I further elaborate that in the revised manuscript:

These results indicate that explore.models is a reliable, valid, and feasible tool to conduct data-driven model exploration with datasets collected from multiple groups in research on health psychology and education. (p. 1)

Comment 6.

Introduction

The introduction includes interesting data from a study regarding COVID-19 vaccination intent across many countries and suggests ways to improve the analysis of this data through more sophisticated models. The author explains that there is a gap for their style of models, identifying other literature that supports the development of their model. Also, the author gives a thorough explanation of the reasoning behind why they designed their study the way they did. But this may be more of a beneficial addition to the Methods section. Despite defining the gap and having some interesting data, major edits must be made to this introduction. One important detail that is missing from this introduction is the objectives of this article and what the researchers’ goals are. A lack of objectives gives the article a lack of purpose.

Response 6.

Thanks for your suggestion to add the aim of the study. Although the aim was already added to the current study subsection, for a better visibility, I added a brief paragraph describing that at the beginning of the introduction:

In the present study, I intend to develop and test a feasible tool to conduct data- driven model exploration within the context of cross-cultural or national research in health psychology and education. This will be done by developing an R package, which is freely available to researchers, implementing fast model searching when a dataset with candidate predictors is given. To examine the performance of this tool, I will test the developed tool with three concrete datasets collected by health psychology researchers in the field. (p. 1)

Comment 7.

Another major flaw of the introduction is that it requires major grammatical revisions. The introduction is hard to follow due to the abundance of run-on sentences and wordy explanations. Additionally, many different concepts are introduced but not thoroughly explained, as if there is an assumption that the reader is already an expert in this subject. Many of the studies and ideas mentioned would benefit from further explanation from the author, so the reader can comprehend why these are relevant to the introduction and the article.

Line 14: Why is there a 1. in front of introduction? There are not any other sections that are numbered, suggest removing this.

Response 7.

Thanks for your comment about the heading. I corrected the LaTeX error throughout the whole manuscript.

Comment 8.

Line 17-20: This is a very long, run-on sentence. Consider breaking up into two sentences, for example. “The mixed-effects model method enables us to examine associations between predictors and the dependent variable of interest at the population level (fixed effects). It also allows us to assess how the intercepts (random intercepts) and the aforementioned associations vary across different groups (random slopes), especially when observations are nested within groups [1]."

Line 18: “(fixed effects)” what does it mean?

Line 19: “(random intercepts)” what does it mean?

Line 20: “(random slopes)” what does it mean?

Line 21-23: “in analysis, analysis” Consider revision of this entire sentence, putting the same word twice is hard for readers to comprehend. Consider the following revision: When data exists in multiple groups, failing to consider group-level factors in analysis can lead to misleading results.

Line 24: so likely resulting in to end up overconfident estimates

Line 25: For instance, we may consider A global public health study across 43 countries conducted in the field of global public health,

Line 26: which explored the relationship between people’s trust in government and science, and

Line 27: COVID-19 vaccine intent across 43 countries [ 4].

Line 28: Replace “get” with a more professional word, such as receive.

Line 30: to be biased.

Line 31: countries, while assuming predictors are the same [6], this possibility warrants warranting the necessity

Line 33: in different across various countries. Consider using another word since “different” was utilized in the Line above.

Line 35: Adding a semi colon makes this sentence too long. Consider ending the sentence in Line 34 and starting the next sentence: “This suggests that…” In fact, the analysis results in

Line 36: regression models, including random intercepts and slopes,

Line 37: significantly better predicted outcome variables significantly better in comparison compared with the simpler models that only

Line 38: possessed with fixed effects [4].

Line 41: “what shall we be supposed to do?” does not make grammatical sense, must be rewritten. It is an interesting way to propose a question in the article to make readers guess what comes next. Consider “we should consider what actions to take next”

Line 43: examined

Line 44-47: This is a very long sentence that is hard to follow. Consider breaking up into two sentences for better flow for readers.

Line 52: might not be able may be unable to accurately predict

Line 53: data accurately [12].

Line 56: Then, If researchers are genuinely interested in searching for the model that best explaining explains

Response 8.

I sincerely appreciate these comments for language correction. I addressed all of them in the revised manuscript. When you required adding additional information about the terms and concepts introduced in the main text (e.g., fixed / random effects), I added such information in the main text.

Comment 9.

Line 58-59: Is the intent of this sentence to lead into the following paragraph, Methods for Model Exploration? If not, consider elaborating on these methods and employed, because this sentence seems like an unrelated ending to the paragraph.

Response 9.

Thanks for your suggestion. I agree with you that the additional context should be provided at that point for a better transition:

In the following subsection, I will overview several existing methods for simple model exploration in the field, such as step-wise regression, Bayesian model exploration, Bayesian Model Averaging (BMA), and regularization. (p. 2)

Comment 10.

Line 62: researchers or the researcher

Line 68-70: Consider combining these two sentences and removing such as forward and backward selection. “First, the variable selection process can be arbitrary, for example, different stepwise methods may suggest different outcome models”

Line 90: to others when given data

Line 93-94: This is not a complete sentence “However, there are several practical limitations warrant further considerations.” Consider adding limitations that warrant                                                   

Lines 96-99: Second, in the case of MLM, which is the main interest of this paper, the existing R libraries, such as BayesFactor, implementing diverse model exploration with feasibility only allowing exploring models with random intercepts, but without random slopes [21].

Line 101: Abbreviations for AIC and BIC should be introduced earlier as they are utilized prior to being defined here.

Line 103: above-mentioned should be edited, for example to mentioned above

Line 117-121: This is a very long sentence, consider breaking into multiple sentences for better flow for the reader

Line 122: In addition, in general,

Line 124: Compared with what criteria? What is the criteria?

Line 125: “despite their calculation is computationally heavier.” is grammatically incorrect. Consider revision, for example despite their heavier computational calculation

Response 10.

I appreciate your comments for language corrections and further clarifications. I revised the parts as you suggested. AIC and BIC were fully defined when they were first used in the manuscript.

Comment 11.

Line 126: Some argued for BIC must be clarified. Who argued for BIC? What was the basis for this argument? More detail is necessary.

Response 11.

Thanks for your comment about the clarification regarding the debates about BIC. In the revised manuscript, I provided additional information:

Another point to consider is that some statisticians, particularly Weakliem [28], argued that BIC assumes the unit prior distribution that might deviate from the actual prior distribution of model probabilities in a specific research project. He suggested that researchers should calculate Bayes Factors with prior distributions that were carefully determined by the researchers [28]. (p. 4)

Comment 12.

Line 135: time,

Line 150: particularly when a complex model is examined Consider removing to improve flow of sentence

Line 157-158: Author refers to Blackburn et al., Han, and Ntontis et al. as if they have been introduced to the reader, but they have not

Line 162: Overuse of the word abovementioned, consider utilizing another word

Response 12.

Thanks for your comments again. I corrected the language issues accordingly. Also, I introduced the three studies briefly along with why they are used in the present study in the main text:

Blackburn et al. [4] examined which factors predict one’s intent to receive COVID- 19 vaccine. Ntontis et al. [35] tested predictors that were significantly associated with psychological well-being during the COVID-19 pandemic. Han [5] explored the relationship between trust in society and compliance with preventive measures during the pandemic. Given all these three datasets were collected across different countries, they are suitable for MLM. (p. 5)

Comment 13.

Methods

The author provides tutorials in a working link at the beginning of the section so readers can utilize the codes for themselves. Also, the author provides very detailed lists and explanations of possible candidate models, while walking the reader step-by-step through their methods. The author addresses a gap in literature with an explanation of how they created this novel model with a detailed tutorial. However, there are major grammatical errors that persist into the methods section and take away from the author’s hard work. These run-on sentences and grammatical errors must be revised in order to allow readers to fully understand the methods section. In conclusion, the reader is unable to follow the methods to replicate the author’s model because the methods section is grammatically flawed.

Line 182-185: Consider revising the structure of this sentence, utilization of semi-colons is excessive

Line 222-224: “When calling explore.models, four parameters, the group variable, the list of random slopes, the list of variables that must be included, and the number of cores to be are not required.” Are all of these elements not required? Or just the number of cores? Consider clarifying and revising this sentence

Line 275: models was 97, indicating that

Line 288: Consider removing (7)

Line 304-305: Consider removing “as well.” It is repetitive after starting the sentence with “In addition to”

Line 305-306: Must consider rewriting the sentence “The processing time was analyzed to examine whether my model exploration method can complete comparing all possible candidate models more quickly than Bayesian MLMs”

Line 308: above-mentioned… We must be consistent with how we are spelling this word, there are times were it is spelled abovementioned and here it is hyphenated. Please edit for consistency

Line 309: Because Bayesian MLMs requires an extremely long time

Response 14.

I sincerely appreciate your invaluable comments and suggestions to improve the methods section. I followed your comments and suggestions to address language issues. Also, I modified multiple “above-mentioned” throughout the manuscript for a better word use.

Comment 15.

Results

The results section contains major flaws. It is recommended to refrain from using I (first person) in scholarly writing. It was noted several times throughout this section. Additionally, the results section should solely focus on the results. The interpretation of the results and their significance should go in the discussion section. Please find the edits for this section below.

Line 316: Instead of I conducted, it is recommended to say  “this research was conducted with”

Response 16.

Thank you very much for your feedback on the results section. Although I followed the most of your comments and suggestions, there is one thing that I would like to clarify. In the manuscript, I used the first-person pronoun intentionally because doing so is recommended and even encouraged in the field of psychology. In fact, the most up-to-date version of the APA guidelines clearly state that first-person expressions should be used in general while minimizing unnecessary uses of third-person expressions and passive voices. I decided to use “I” throughout the manuscript following such official guidelines to write academic papers in the field of psychology. For further information, please refer to these documents:

https://owl.purdue.edu/owl/research_and_citation/apa6_style/apa_formatting_and_style_guide/apa_stylistics_basics.html

https://apastyle.apa.org/style-grammar-guidelines/grammar/first-person-pronouns

https://apastyle.apa.org/blog/first-person-myth

Other than this, I followed your comments and suggestions to improve language in the results section.

Comment 17.

Line 317: “with explore.models with models suggested by explore.models”  Using with twice in a row sounds redundant,  also said models frequently in this sentence

Line 326-327: “Only included one random slope, primary stressors” It is suggested to say “only included one random slope, which was primary stressors”

Line 332: Instead of suggesting , use “suggests”

Line 335: It is recommend to say which was similar to after the word predictors

Line 337: There should not be a comma after the and, also suggest deleting the word and

Line 339: “When the BIC-best model as” Use was instead of as

Line 343: “,10” It is recommended to say which was 10

Line 343: “Thus, I shall conclude that the models” à “It can be concluded that the models”

Line 346: “was not significant, 2(4297.25 − 4296.58) = 1.34” It is suggested to say was 2(4297.25 − 4296.58) = 1.34 and not significant

Line 347: Instead of which, start new sentence with this was below

Line 351: Instead of saying as expected, say “as predicted”

Line 353: “One note is that when” it is recommended to say of note, when Blackburn

Line 359: “Even if we assume” it is recommended to say even if one assumes

Line 362: “,the actual processing time would be” It is suggested to say this means the actual processing time

Line 363: “estimate, 43637.75 seconds” instead of the comma, say estimate of 43637.75 seconds

Line 363: Do not say as expected, it is suggested to say “similar to the predictions of this study”

Line 364: Suggest taking out “the” before the word shorter

Line 365: “The same trends was also found from my analysis of” Instead of was, use were. Also instead of saying my, say “found from the analysis of”

Response 18.

Thank you very much for your suggestions. I corrected these errors in the manuscript. Additional changes were made based on feedback provided by a professional language tool.

Comment 19.

Line 363-366: This part of the results likely belongs in the discussion section.

Response 19.

Thanks for your suggestion. I moved this part to the discussion section:

In addition, when compared with Bayesian MLM, explore.models could explore and compare all possible candidate models, which ranged from 13 ([4]) to 2315 ([5]) models, within a significantly shorter processing time. Similar to the prediction of this study, explore.models was found to require a shorter processing time to test all possible candidate models than Bayesian MLM. The same trends were also found from the analysis of Blackburn et al. and Ntontis et al. [4,35]. The nominated models reported better BFs compared with the full models. Hence, the nominated stringent models are less susceptible to over-fitting as reported by better cross-validation results from previous studies (e.g., Han et al. [13,14]). Researchers who intend to conduct data-driven model exploration with multilevel models can use explore.models to save time, while also maintaining the credibility of the model selection process. The R codes include customized functions and tutorials available to the public via GitHub, meaning researchers will be able to feasibly employ explore.models in their research projects. (p. 10)

Comment 20.

Discussion:

The discussion succeeded in comparing the model to other models and explaining the use for the model. The limitations were noted well with specific examples. Since the paper did not include a conclusion, some of the text from the discussion section should be added to the conclusion. The conclusion should restate the main points and discuss next steps. A weakness of this paper was the lack of further direction for future studies. Please find the edits for this section below.

Line 369-371: “explore.models, which I invented, allows its users to explore the best prediction model among candidate models; the candidate models are generated by the combination of candidate predictors at the population and group levels following the users’ directions.”

  • The sentence should not start with Explore.models

  • Which I inventedà It is recommended to say “invented by the author”, want to refrain from using first person

  • Instead of a semicolon highlighted in red, it is suggested to put a period and make the second part a different sentence.

Line 373: outcomes and to inflate false positivesà It is suggested to delete the word “to”

Response 20.

I appreciate your feedback on the discussion section for language improvements. I revised the manuscript accordingly while considering feedback provided by a professional language tool as well.

Comment 21.

Line 380: Should the E be capital in explore.Models because it is the start of a sentence?

Response 21.

Thanks for your comment about the word. This should be remained as it is since it is an official name of the function in the developed R package. Function names in R are case sensitive, so capitalizing the first letter will make the function name unrecognizable by R.

Comment 22.

Line 381: What is brms? It is not defined anywhere else in the paper. 

Response 22.

Thank you for your request to specify brms. It is an R package to conduct Bayesian MLM. I updated the methods section to mention the role of brms the current study:

For functionality testing, I conducted Bayesian MLM with brms [44,45]. Bayesian MLM was performed with the best models identified in terms of the lowest BIC and AIC, the null model only including designated demographic variables, and the full model including all candidate predictors and random effects. (p. 8)

Comment 23.

Line 385: “Stringent models are less likely susceptible” Recommend saying either “less susceptible” or “less likely to be susceptible”

Line 387-389: “In general, I found that researchers who intend to conduct data-driven model exploration with multilevel models will be able to employ explore.models to save their time while maintaining the credibility of the model selection process.”

  • Rewording suggestion: Researchers who intend to conduct data-driven model exploration with multilevel models can use explore.models to save time, while also maintaining credibility of the model selection process.

Lines 389-390: “Also, since I composed the R codes, which include the customized functions, and tutorials available to the public via GitHub, researchers will be able to employ explore.models in their research projects feasibly.”

  • Rewording suggestion: The R codes include customized functions and tutorials available to the public via GitHub, meaning researchers will be able to feasibly employ explore.models in their research projects.

Line 393: “information criteria are” Say “is” in the place of “are”

Line 397: “been regarded practically acceptable” Rewording suggestion: Has been regarded as acceptable

Line 398: “with proper prior distributions coherent with data” Consider rewording

Line 399: Recommend changing “that suggests” to “this suggests”

Line 402-403: “It might allow researchers to compromise between computational complexity and credibility of model recommendation reasonably.” Suggest changing to: It could allow researchers to reasonably compromise between computational complexity and credibility of model recommendation.

Line 404-406: “Second, although I was able to boost the processing time via use of information criteria and multiprocessing, it still requires a long time to complete model exploration with a complex model.” Rewording suggestion: Although this study was able to boost the processing time via use of information criteria and multiprocessing, it still requires a lengthy amount of time to complete model exploration with a complex model.

Line 407-408: “13 models with 2  predictors → 97 models with 4 predictors → 2315 models with 7 predictors” Recommend writing this out in a sentence instead of using arrows.         

Line 409: “3.06 seconds → 18.83 seconds 409 → 3314.29 seconds” Recommend writing this out in a sentence instead of using arrows.         

Line 411-412: “setting more restrictions (e.g., specifying more "should be 411 included" variables or less candidate random slopes” It is suggested to say “setting more restrictions, such as”

Line 414: “ Third, although” It is recommended to say Even though instead

Line 415-416: “the models do not necessarily provide theoretically and conceptually relevant and meaningful results” Rewording suggestion: Theoretically and conceptually relevant results that are meaningful.

Line 418: Possibly à This word can be deleted

Line 420: “Also, researchers will need to” à Rewording suggestion: Researchers will also need to consider

Line 423-424: “spending an unnecessarily long time for computation.” Rewording suggestion: spending an excessive amount of time computing the data

Line 424: “they may consider” It is suggested to say “they must consider”

Line 426: Instead of (e.g.,) Recommend making this a separate sentence and start it with for example

Line 430: Used also twice in this sentence, suggest deleting the also at the end of the sentence

Line 430: Apparently dataà Is this a term used in the field? The sentence read weird due to the word apparently.

Line 439: Do not start a sentence of with of course, you could delete this and start with the word ideally.

Response 23.

I sincerely appreciate your additional comments and suggestions. I addressed all of these points in the revised manuscript.

Comment 24.

Conclusion: This article did not have a conclusion section; author should add one at the end of the discussion section.

Response 24.

Thank you very much for your suggestion. I created a short conclusions section to briefly summarize the current study. Plus, at the end of the discussion section, I briefly discussed the future directions to address limitations:

Perhaps, additional studies should be conducted to optimize the computational process and to develop guidelines to determine prior distributions and which criterion should be employed for model recommendation.

  1. Conclusions

In the present study, I developed and tested an R tool for model exploration for mixed- effect analysis. The developed tool, i.e., explore.models, could generate a credible list of candidate models for further exploration while minimizing processing time. I shared the source codes and tutorial files via GitHub, so researchers in the field of health psychology and education, particularly those who intend to conduct cross-cultural and cross-national research projects, will be able to employ this tool for data-driven analysis in their research project feasibly. (p. 12)

Reviewer 4 Report

Comments and Suggestions for Authors

This paper describes the R module 'explore.models' which can explore best models within the context of multilevel modeling in research in public health. It is a clear and well-written paper, but I have some suggestions for improvement.

Major:

- Since this publication is submitted to the "European Journal of Investigation in Health, Psychology and Education" and not a statistics journal, I believe the first part of the introduction, which introduces mixed-effects models, should be expanded. Also give a more detailed explanation on how mixed-effects models are used in this field.

- In some countries, people tend not be vaccinated by religious beliefs. In these cases they might still trust the vaccine to work, but they believe that it's not morally acceptable. See e.g. https://www.ncbi.nlm.nih.gov/pmc/articles/PMC5141457/ for more background. This should be discussed and perhaps included in the analysis, e.g. in formula 6.

Minor:

- Line 95: "MLM" -> "multilevel model (MLM)"

- I think the data availability statement should also include the OSF link from line 290.

Comments on the Quality of English Language

- Abstract: "The module ... compare" -> "The module ... compares"

- Formula 6: "beliefes" -> "beliefs"

- Line 304:"also ... as well" (double)

Author Response

Dear Reviewer 4,

Thank you very much for your invaluable comments to improve my paper. You can find how to revised the manuscript as per your comments below:

Comment 1.

  • Since this publication is submitted to the "European Journal of Investigation in Health, Psychology and Education" and not a statistics journal, I believe the first part of the introduction, which introduces mixed-effects models, should be expanded. Also give a more detailed explanation on how mixed-effects models are used in this field.

Response 1.

Thank you very much for your suggestion to connect the current study and the journal's context. In the revised manuscript, I added additional information about why this study can be a significant contribution to the field by referring papers using MLM published in the EJIHPE:

Furthermore, many health psychology and education researchers interested in address- ing their research questions across multiple countries or contexts have widely employed mixed-effect analysis methods. For example, we can find several studies using this ap- proach from the European Journal of Investigation in Health, Psychology and Education. Nasvytiene ̇ and Lazdauskas [8], Ta et al. [9], and Lochbaum and Sisneros [10] included both fixed-effects and random-effects in their analysis models to examine the relationship between candidate predictors and outcome variables across different conditions and con- texts in the field of health psychology and education. As demonstrated by these papers, mixed-effect analysis methods have been frequently used in research on health psychology and education beyond COVID-19-related research. (p. 2)

Comment 2.

  • In some countries, people tend not be vaccinated by religious beliefs. In these cases they might still trust the vaccine to work, but they believe that it's not morally acceptable. See e.g. https://www.ncbi.nlm.nih.gov/pmc/articles/PMC5141457/ for more background. This should be discussed and perhaps included in the analysis, e.g. in formula 6.

Response 2.

Thanks for your comment about religiosity and vaccine intent. I briefly discussed that point and the limitation of the previous study in the revised manuscript:

One minor point for consideration is that this study did not examine one’s religious belief as a predictor [43]. Given some people may not want to get vaccinated due to their religious beliefs even if they trust the effectiveness of vaccines, the analyzed dataset might not include all potential predictors comprehensively. Thus, this might need to be noted as a potential limitation of the analyzed dataset. (p. 7)

Comment 3.

Minor:

- Line 95: "MLM" -> "multilevel model (MLM)"

- I think the data availability statement should also include the OSF link from line 290.

- Abstract: "The module ... compare" -> "The module ... compares"

- Formula 6: "beliefes" -> "beliefs"

  • Line 304:"also ... as well" (double)

Response 3.

I sincerely appreciate your minor comments to further improve the language and clarity of the manuscript. While revising the manuscript, I addressed the issues that you raised. Plus, I employed a professional tool to correct language issues in the paper. The OSF link was added to the data statement accordingly.

Round 2

Reviewer 3 Report

Comments and Suggestions for Authors

The article introduces a creative new model for multiprocessing public health research. A gap exists for this style of model, which could be filled by the specific model described in the article. Despite the innovative approach and the potential of the model described within, the presence of numerous grammatical errors and flaws decrease the quality of the article. For instance, the introduction lacks clarity on the importance of the model and its historical context within public health research. Moreover, despite suggested revisions, grammatical errors persist throughout the article, detracting from its overall quality and coherence. For example, in Lines 41-42, the content, “A global public health study across 43 countries conducted in the field of global public health” must be revised again.

Additionally, the introduction starts with an explanation of the study, which should be in the methods.

Second, the results section was difficult to comprehend, and included some of the methodology, such as “Then, I calculated model BFs, BFAIC,0, BFBIC,0, and BFFull,0 375 to examine the extent to which each model was supported by evidence.” Although the author fully explained the limitations of this study, such as the model being time consuming, this might lower the significance and use of the model.

Overall, the article was hard to understand, and the model may have limited real world use.

Furthermore, additional clarification is needed in certain areas to ensure that researchers can readily interpret the study findings and effectively utilize the model in practice.

In conclusion, this article still requires major edits after revision, and is not acceptable for publication.

Comments on the Quality of English Language

 Lines 41-42, the content, “A global public health study across 43 countries conducted in the field of global public health” must be revised again.

"Then, I calculated model BFs, BFAIC,0, BFBIC,0, and BFFull,0 375 to examine the extent to which each model was supported by evidence.” needs to be addressed.

Author Response

Dear Reviewer 3,

First of all, thank you very much for your time and effort to provide me with your invaluable feedback on the revised manuscript. Attached please find the manuscript revised once again as per your comments and suggestions. You can find my responses to your comments below.

Thanks a lot for your time and consideration once again. I hope you will find improvements from the updated version.

Sincerely,

Hyemin Han

-----

Comment 1.

The article introduces a creative new model for multiprocessing public health research. A gap exists for this style of model, which could be filled by the specific model described in the article. Despite the innovative approach and the potential of the model described within, the presence of numerous grammatical errors and flaws decrease the quality of the article. For instance, the introduction lacks clarity on the importance of the model and its historical context within public health research. Moreover, despite suggested revisions, grammatical errors persist throughout the article, detracting from its overall quality and coherence. For example, in Lines 41-42, the content, “A global public health study across 43 countries conducted in the field of global public health” must be revised again.

Response 1

Thank you very much for your comment about the language of the revised manuscript. I agree with you that there are still several places requiring further editing. To improve the quality of the language, the whole manuscript was reviewed and edited once again with a professional language editing tool. This time, I focused on not only obvious grammar issues but also the overall clarity and fluency. Given the attached manuscript PDF shows which parts have been updated and modified, you will be able to find the places edited and corrected.

Comment 2.

Additionally, the introduction starts with an explanation of the study, which should be in the methods.

Response 2.

I appreciate your comment about the first paragraph in the introduction. That was added during the first round of the revision process based on another reviewer's suggestion. I agree with you that that is not the ideal place to locate such a paragraph. In the revised manuscript, the paragraph was removed from the introduction.

Comment 3.

Second, the results section was difficult to comprehend, and included some of the methodology, such as “Then, I calculated model BFs, BFAIC,0, BFBIC,0, and BFFull,0 375 to examine the extent to which each model was supported by evidence.” 

Response 3.

Thanks for your comment about the results reported in the manuscript. During the current revision process, I modified the parts where results were not clearly reported to be easily read and understood by readers. For instance, the part that you mentioned was updated with multiple separate sentences:

Table 1 demonstrates the result of model exploration with three datasets. I conducted Bayesian MLM with models suggested by explore.models, which were based on AIC and BIC, the full and null models. Then, I calculated model BFs. First, BFAIC,0 indicates the extent to which the AIC-best model was more strongly supported by evidence than the null model. Second, BFBIC,0 provides information about the extent to which the BIC-best model was more favored by data than the null model. Finally, BFFull,0 was calculated to examine the extent to which the full model including all candidate predictors was more strongly
supported by evidence than the null model. In the case of the simplest model exploration, Blackburn et al. [4], the AIC- and BIC-best models were identical to the full model, so only one model BF value was reported. (p. 8)

Comment 4.

Although the author fully explained the limitations of this study, such as the model being time consuming, this might lower the significance and use of the model.

Response 4.

I appreciate your comment about the potential impact and significant of my study. In the revised manuscript, in the part in the limitations section addressing the processing time issue, I added a statement that employing multiprocessing will significantly improve the performance of the tool so that users will be able to use it feasibly even with limited time and resources. I presented such a point with a concrete study examining the impact of multiprocessing on processing time in academic cluster computing:

Alternatively, researchers may employ large-scale cluster computing to increase the number of cores for calculation [ 23 ]. Han [23 ] demonstrated that utilizing 16 cores, which is feasibly done via available computer clusters, such as Amazon AWS, decreased processing time by more than 90%. Given I employed only four cores in the present study, cluster computing will effectively reduce processing time even while examining complex models and make the current method more feasible for researchers with limited time and resources. (p. 12)

Comment 5.

Overall, the article was hard to understand, and the model may have limited real world use.

Furthermore, additional clarification is needed in certain areas to ensure that researchers can readily interpret the study findings and effectively utilize the model in practice.

In conclusion, this article still requires major edits after revision, and is not acceptable for publication.

Response 5.

Thanks for the potential impact and significance of the manuscript. As the manuscript has been throughly reviewed and edited once again while examining additional features, I hope you will find how it has been improved from the previous version. If additional edit(s) are still needed after the current revision, I will employ another tool or service to conduct language edit from another perspective once again.

I also hope that the potential significance and impact of the current work, which has been considered as a potential limitation from your perspective, will be well addressed via a suggestion regarding use of multiprocessing. Given cluster computing is not very expensive and feasibly implemented by researchers these days, the weak point of the tool regarding relatively long processing to explore complex models might not be a serious issue in real applications.

Round 3

Reviewer 3 Report

Comments and Suggestions for Authors I would like to accept this manuscript, which I hope will be rigorously reviewed for grammar and sentence structure. Comments on the Quality of English Language I hope will be rigorously reviewed for grammar and sentence structure.

Author Response

Dear Reviewer 3,

Thank you very much for your comment and suggestion once again. I double checked the quality of the language in my paper with the assistance of another professional language editing tool. I hope you will find significant improvements. Also, I expect that the MDPI journal office will take another look at the manuscript with fresh eyes to correct remaining errors and issues.

Best regards,

Hyemin Han